# Deep convective cloud system size and structure across the global tropics and subtropics

Eric M. Wilcox[1], Tianle Yuan[2,3], Hua Song[3,4]

[1]Division of Atmospheric Sciences, Desert Research Institute, 2215 Raggio Parkway, Reno, NV, 89512, USA
5    [2]Joint Center for Earth Systems Technology, University of Maryland, Baltimore County, Baltimore, MD, USA
[3]Sciences and Exploration Directorate, Goddard Space Flight Center, Greenbelt, MD, USA.
[4]SSAI Inc., Lanham, MD, USA

September 14, 2023

10    *Correspondence to*: Eric M. Wilcox (Eric.Wilcox@dri.edu)

**Abstract.** A new database is constructed comprising millions of deep convective clouds that spans the global tropics and subtropics for the entire record of the MODIS instruments on the Terra and Aqua satellites. The database is a collection of individual cloud objects ranging from isolated convective cells to mesoscale convective cloud systems spanning hundreds of thousands of square kilometers in cloud area. By matching clouds in the database with the MERRA-2 reanalysis dataset and 15    microwave imager brightness temperatures from the AMSR-E instrument, the database is designed to explore the relationships among the horizontal scale of cloud systems, the thermodynamic environment within which the cloud resides, the amount of aerosol in the environment, and indicators of the microphysical structure of the clouds. We find that the maximum values of convective available potential energy and vertical shear of the horizontal wind associated with a cloud impose a strong constraint on the size attained by convective cloud system, although the relationship varies geographically. The cloud database 20    provides a means of empirical study of the factors that determine the spatial structure and coverage of convective cloud systems, which are strongly related to the overall radiative forcing by cloud systems. The observed relationships between cloud system size and structure from this database can be compared with similar relationships derived from simulated clouds in atmospheric models to evaluate the representation of clouds and convection in weather forecast and climate projection simulations, including whether models exhibit the same relationships between the atmospheric environment and cloud system 25    size and structure. Furthermore, the dataset is designed to probe the impacts of aerosols on the size and structure of deep convective cloud systems.

## 1 Introduction

Deep convective clouds span a range of scales from individual convective cells on the order of 1 kilometer or smaller in diameter to giant mesoscale convective clusters spanning many hundreds or even thousands of kilometers in diameter. Such 30    clouds form in regions where a conditionally unstable environment exists in a deep layer of the troposphere. These are conditions that are commonplace in the tropics, sometimes present in the subtropics, and occur episodically at higher latitudes, such as over summertime midlatitude continental regions.

Convection is a central feature of the general circulation, providing the vertical fluxes of mass, momentum and energy required to balance the destabilizing influence of the net radiative fluxes. The cloud systems themselves feedback on the thermodynamic

structure of their environment through latent heat of condensation and fusion, as well as cloud radiative forcing. The horizontal scales of cloud systems are important because it is the largest mesoscale convective systems that dominate the total contribution of clouds to latent heating and cloud radiative forcing in the tropics (Wilcox and Ramanathan 2001). For these reasons, as well as their contribution to extreme precipitation events, it is essential to understand the processes that lead to the growth of these large systems.

The representation of convection and convective clouds in atmospheric models has been a long-standing challenge in the simulation of climate. Limitations in fidelity of convection and cloud parameterizations have contributed substantial uncertainty in simulations of the general circulation (Bechtold et al. 2008; Oueslati and Bellon 2013; Becker et al. 2017), cloud distributions and feedbacks to increasing greenhouse gases (Sherwood et al. 2014; Tomassini et al. 2015), and changes in precipitation, such as extreme rain events (Wilcox and Donner 2007; Bush et al. 2014). These shortcomings arise from a poor

understanding of the processes that cause convective cloud systems to grow and a lack of representation in cloud models of the processes that cause convection to organize into large convective clusters (Wilcox 2003; Lin et al. 2006 and Bony et al. 2015).

Here we present a new database of millions of convective clouds intended to support weather and climate studies of convective cloud process by linking the size and structure of convective cloud systems to aspects of their environment in which they are

forming, and thus providing, among other benefits, an empirical means of evaluating models and parameterizations of convective cloud systems. The database is constructed using infrared brightness temperatures from MODIS to identify the boundaries of convective clouds. An objective algorithm associates anvil cloudiness with convective cores and attaches the cores and anvils within the same convective cloud object. These cloud objects are then evaluated to quantify their size and structure and relate these properties to environmental conditions in which the cloud resides, including aspects of the

thermodynamic structure of the atmosphere and the aerosol optical thickness in the vicinity of the cloud.

In what follows we provide a detailed description of the methodology upon which the cloud database is constructed, review some of the climatological aspects of the resulting collection of clouds, and propose further applications of the database, including the evaluation of the representation of clouds in weather and climate models, and exploring the impacts of aerosols on the microphysical and macrophysical properties of clouds.

**2. Constructing a cloud database**

To construct a database of cloud objects, we seek to identify individual objects that represent single unique elements of the dynamical feature that defines deep convective clouds. At the smaller end of the convective scale these clouds are defined by a single cell of vertically-oriented overturning motion, where the cloud resides in the rising portion of the circulation. At the larger end of the size scale, the mesoscale, convective clouds systems are composed of convective cores attached to stratiform

anvil cloud. A single such convective cloud structure can have many convective cores embedded within a single large stratiform cloud shield. We apply an objective algorithm designed to identify as unique elements both isolated convective cells and mesoscale convective systems.

    Once the satellite pixels associated with individual clouds are identified, we co-locate the clouds with ancillary information about the environment in which the cloud has formed derived from other MODIS products and NASA MERRA-2 reanalysis

products derived from assimilating MODIS and other satellite data in a dynamical model of the atmosphere.

    The fucus of this cloud database product has been the application of greater than 20 years of unique observations from polar-orbiting satellite systems that go beyond just infrared brightness temperatures to include co-located microwave brightness temperatures and geophysical retrievals of quantities including the cloud droplet particle effective radius. A limitation of these data, however, is that the polar orbiting nature of the satellite systems means that we cannot track clouds through time and

characterize how the structural elements of cloud clouds observed here are evolving through the lifecycle of the convective cloud. In the discussion that follows we note where such a characterization would add considerable further value, and in section 5 below we address further development of the dataset to address this limitation.

## 2.1 Application of the Detect and Spread algorithm to infrared brightness temperatures

    We apply an iterative two-step algorithm to achieve the goal of objectively identifying distinct deep convective cloud objects

in an image of a cloudy scene such as those produced from satellite imaging instruments. The algorithm, referred to as Detect and Spread, was originally described by Boer and Ramanathan (1997). They describe a generalized approach where a pixel identified as cloudy in an image is connected to adjacent cloudy pixels and given identical labels that associate the adjacent pixels as part of the same cloud object. When there are no more pixels identified as cloudy that are adjacent to any of the pixels already labelled as part of the cloud object, the algorithm identifies a new un-labelled cloudy pixel and performs the same

process but applies a new label that identifies the new adjacent cloudy pixels as a different cloud object.

    Cloud object labelling as described above pre-dates the Boer and Ramanathan (1997) study. For example, Mapes and Houze (1993) provide an earlier example and include a review of other pioneering efforts using infrared satellite imagery. In the novel approach described by Boer and Ramanathan (1997) this labelling is achieved with an iterative process where new seeds (i.e. the detection of new un-labelled cloudy pixels) are initially identified using a relatively cold threshold in infrared brightness

temperature, indicating deep convection, and the attaching of adjacent cloudy pixels (i.e. the spreading process) is based on a slightly warmer threshold. This two-step process is repeated in multiple iterations with Detect and Spread thresholds that increase in temperature and thereby approach a clear-sky condition.

    In this manner, the coldest portion of the cloud is identified as the core of the convective motion and warmer adjacent cloudiness is attached to the core through the spreading process. To minimize the possibility that a cloud is spread to encompass

a separate neighboring cloud, the algorithm is applied in multiple Detect and Spread iterations each over a limited range of brightness temperatures. Furthermore, each detect step applies a temperature threshold that is slightly cooler than previous spread step.

The progression of the multiple iterations of Detect and Spread is illustrated in Fig. 1, which shows and example of the algorithm as applied to a single granule of the MODIS-Aqua observations over South America from 27 Jan. 2011, 1720 UTC.

At the top of the figure is the infrared brightness temperature image from which the clouds are identified. In the second row, left is an image illustrating the result of the first detect step, based on a threshold of 220 K, where each contiguous region of the same color is a convective core region of an individual cloud object. In the second row, right is the same image after the first spread step has grown the initial core regions out to a warmer threshold of 240 K. The third row of images in Fig. 1 shows the result after the addition of a second detect step at a threshold of 235 K on the left, and on the right the result after all of the

clouds identified so far are spread to a threshold of 255 K. The single image on the bottom row shows the result after the addition of a final detect step at a threshold of 255 K.

Boer and Ramanathan (1997) choose values for the infrared thresholds designed to label every cloudy pixel in a series of satellite images over the western Pacific Ocean where after multiple Detect and Spread iterations cloud objects are spread to a threshold of 285 K chosen to discriminate cloudy pixels from clear-sky pixels. For the dataset presented here we adopt the

application of Detect and Spread in Roca and Ramanathan (2000) which seeks to identify only the high clouds associated with deep convection and thereby screens out lower clouds or thin cirrus clouds exhibiting brightness temperatures warmer than 255 K.

Detect and Spread was designed to partly address challenges of adjacent clouds separated by a narrow region of optically thin cloud or partially filled pixels (Coakley and Bretherton 1982). It was proposed as an advance on prior methods based on

contours of a single threshold for infrared brightness temperature or visible reflectance and toward a goal of defining cloud boundaries that more closely resemble the true boundaries of clouds (Boer and Ramanathan 1997).

In the cloud database constructed here we apply Detect and Spread to the infrared window channel (11 μm) brightness temperatures from MODIS. The native pixel size of the MODIS 5-minute granule product files (MOD021KM) is nominally 1 km at nadir, but the data are first re-gridded to a coarser resolution for purposes of collocating with the Advanced Microwave

Scanning Radiometer-EOS (AMSR-E) 89 GHz data as described further in Sect. 2.2 below. The sensitivity of the cloud size statistics to the choice of grid size for this re-gridding is examined in Sect. 4 below.

## 2.2 Co-location of 89 GHz polarization corrected brightness temperature

While infrared brightness temperatures are used here to discriminate the boundaries of cloud objects, an alternative that has also been applied to deep convective clouds is to use the 89 GHz polarization corrected brightness temperature (PCT) from

passive microwave imagery, such as from the AMSR-E instrument (Wentz and Meissner, 2000; Wentz et al., 2003). An example of this (based on similar 85 GHz measurements) is the dataset of cloud objects produced from the TRMM Microwave Imager instrument (Liu et al. 2008). In deep convective clouds the PCT is sensitive to the presence of precipitation sized ice particles, which are predominantly hail and graupel (Spencer et al. 1989). Thus, the objects that arise from applying a threshold to this measurement, as in Liu et al. (2008) captures individual features characterizing the convective core and precipitating

anvil portions of the convective cloud. Here, we seek to augment our infrared-based cloud object database by including

statistics derived from the 89 GHz PCT. This is achieved by first gridding the 89 GHz brightness temperatures on the same grid upon which the infrared brightness temperatures have been gridded. Because the pixel size of the 89 GHz AMSR-E brightness temperatures is larger than that of the MODIS infrared brightness temperatures, the common grid is chosen to be 0.125° cell spacing in latitude and longitude. This is approximately 14 km grid cells which is chosen to be larger than the nominal 6 km pixel size of the 89 GHz AMSR-E data. We then include among the cloud statistics in the final dataset the location of the minimum 89 GHz PCT, which coincides with the strongest signature of scattering by precipitation-sized hail and graupel, and the average of all samples of the 89 GHz PCT within the boundary of the cloud. We also include the 89 GHz PCT at the location of the minimum infrared brightness temperature, and the infrared brightness temperature at the location of the minimum 89 GHz PCT. Using these additional statistics it is possible to better explore the relationships between ice scattering characteristics, which generally indicate colder 89 GHz PCT values at locations with efficient ice scattering by hail and graupel resulting from strong vertical motions in the cloud, and the cloud-top temperature distribution, which indicates where the cloud is penetrating deeper into the troposphere or tropopause region.

The AMSR-E instrument was operational from the start of the Aqua mission in mid-2003 until the instrument failed in October 2011. The Aqua cloud database extends beyond 2011 but does not include the 89 GHz PCT values beyond October 2011. There is no collocated microwave imager data for the Terra MODIS observations, therefore the Terra cloud database also does not include the 89 GHz PCT values.

## 2.3 Co-location of environmental properties from MODIS and the MERRA-2 reanalysis

This deep convective cloud database has been created, in part, to explore the relationships among the size and structure of deep convective cloud systems and the environment in which the cloud has developed. In particular, we are interested in investigating aspects of the thermodynamic profile of the atmosphere and the aerosol load in the vicinity of the cloud. This is achieved by collocating the clouds in the cloud database with MODIS gridded daily aerosol optical thickness (AOT) data (MOD08_D3 product) and values of convective available potential energy (CAPE) and vertical shear of the horizontal wind derived from the MERRA-2 reanalysis. Since the MODIS AOT is not retrieved in MODIS samples that are cloudy, the gridded level 3 AOT from MODIS is representative only of the AOT in the vicinity of the cloud. For comparison, we also match clouds with the AOT from the MERRA-2 reanalysis, which includes an aerosol model that is constrained by the clear-sky MODIS AOT data.

Multiple methods of collocation are employed to match clouds with these gridded data. First, we average the gridded fields over the area of clouds. We also evaluate the maximum value of the CAPE and shear at the native grid cell size of MERRA-2 within the boundaries of the cloud. Finally, we evaluate the values of CAPE, shear and AOT at the locations of the minimum IR and minimum 89 GHz PCT within the cloud. Note that clouds amounting to only one or a few of the fine grid cells that the clouds are defined from (0.125° lat/lon, as described in Sect. 2.2 above) will be smaller than the native grid cell size of the level 3 MODIS data and the MERRA-2 reanalysis datasets. The native grid cell size of the MODIS aerosol optical thickness is 1° lat/lon. The native grid cell size of the MERRA-2 reanalysis is 0.5° lat. and 0.625° lon. For the smallest clouds, therefore,

the gridded environment variables from MODIS level 3 products and MERRA-2 reanalysis products represent scales that are larger than the clouds themselves, and the different collocation approaches all result in the same value for the small clouds.

We note that the representation of convection in MERRA-2, though constrained by available observations, is subject to the imperfect parameterization of convection. Hence, it may be the case that for the convective clouds observed by MODIS there will be varying levels of correspondence between an observation of deep convection and the presence of simulated convection in the model. There may indeed be cases where an observed deep convective cloud does not correspond to a simulated deep convective event. The opposite may also be true as well. As shown in section 4.2 below, there is an observed systematic relationship between the size of observed deep convective clouds and the maximum values of the MERRA-2 CAPE and shear. The use of the maximum values of CAPE and shear in this assessment allows for some misalignment of the observed and simulated convection, which is increasingly effective as the cloud system size increases.

To further evaluate the correspondence between MODIS-observed and MERRA-2-simulated convection we present examples of co-located maps of the MODIS infrared brightness and MERRA-2 outgoing longwave radiation in Supplementary Fig. 1. These examples indicate both that often there is some correspondence between the presence of convection in MERRA-2 with the patterns of cloudiness evident in the MODIS imagery, but that the correspondence is certainly imperfect. We note also, that even for a perfect correspondence between the spatio-temporal distribution of convection, we would not necessarily expect a perfect correlation between infrared brightness temperature and outgoing longwave radiation. Of interest here is the spatial patterns of the convection, which correspond to varying degrees between the examples shown. To further characterize the level of correspondence, we examined the level of correlation between the minimum values of MODIS infrared brightness temperature within the boundaries of the clouds observed in these examples with the minimum values of the MERRA-2 outgoing longwave radiation (Supp. Fig. 2). This correlation is examined for all clouds as well as three different thresholds isolating only the larger clouds in this sample population of clouds. There appear to be some outliers where minimum infrared brightness temperature is quite cold (<220 K), but MERRA-2 minimum outgoing longwave radiation is relatively high (>200 – 220 W m-2). These outliers suggest that these may be cases where MODIS is observing a very cold deep convective cloud top, but the MERRA-2 outgoing longwave radiation is not indicating intense convection in MERRA-2. For MODIS-observed clouds with relatively warm minimum infrared brightness temperatures, which are generally the smaller and less intense convective events, there is a broad spectrum of MERRA-2 OLR values.

In the absence of direct global observations of CAPE and shear in the presence of deep convective clouds systems, these results, and those shown in section 4.2 below, suggest that there is some level of correspondence between observed convective cloud systems and MERRA-2 simulations of convection. Nevertheless, there is some uncertainty in the results shown in section 4.2 below due to errors in this correspondence. Greater characterization of this uncertainty will be an ongoing element of research with the cloud database and metrics derived from the MERRA-2 outgoing longwave radiation will be added to a subsequent version of the cloud database in order to improve the characterization of this uncertainty with the full population of clouds in the database along the lines illustrated here in the supplementary figures.

## 2.4 Cloud-top distribution of droplet effective radius

For each cloud in the database, we evaluate the distribution of the cloud hydrometeor effective radius retrieved from all of the cloud top samples within the cloud boundary against the infrared brightness temperature. This approach was first demonstrated
in Rosenfeld and Woodley (2003), where they interpreted the result as a composite profile of the variation of the droplet effective radius with height within the cloud. The result has been shown to be useful for estimating the glaciation level, i.e. the brightness temperature at which the cloud is fully glaciated, in Yuan et al. (2010). In the implementation used here, we utilize all 1 km infrared brightness temperature and effective radius samples residing within the boundaries of a cloud. The effective radius samples are then sorted and averaged in infrared brightness temperature bins of 2 K width from 180 K to 270 K. Note
that while the warmest threshold used in the Detect and Spread to define cloud boundaries is 255 K, it is possible for there to be 1 km samples warmer than 255 K within larger 0.125° grid cells that are equal to or colder than 255 K.

Two examples of the cloud-top hydrometeor distributions included in the cloud database are shown in Fig. 2. Shown are the averages of the profiles from two clusters of approximately 25 clouds each from the same granule as in Fig. 1 over South America from 27 Jan 2011, 1720 UTC. The inferences derived from these distributions are based on a conceptual model for
the development of convective clouds, as described in the references above. In this conceptual model, the cloud drop effective radius generally increases with height in the liquid and mixed phase portions of the cloud, but begins to decrease with height above the level where the cloud becomes fully glaciated. The examples shown here indicate two clusters of clouds where the southern cluster (indicated in the green-colored profile in Fig. 2a and the clouds encircled in green in Figs. 2b and c) exhibit a warmer glaciation temperature, approximately 250 K, compared to the northern cluster (indicated in the blue-colored profile
and the clouds encircled in blue in Fig. 2), which exhibit a glaciation temperature of approximately 240 K.

The assumption implicit in these composite profiles is that the vertical profiles of individual cloud elements in a large cloud system are similar, and therefore we can use cloud tops at varying heights within the cloud system as a proxy for the average profile of the cloud elements. See Rosenfeld and Lensky (1998), Lensky and Rosenfeld (2006), and Yuan et al. (2010) for further discussion and evaluation of this assumption. Note also, that there is likely an inherent evolution of these distributions
through the lifecycle of deep convective clouds that is not yet characterized in this database. The goal of including these cloud-top effective radius distributions in the cloud database is to elucidate the sensitivity of these distributions, as well as the derived glaciation height, to geographic variations in cloud, to land/ocean contrasts, and to potential effects of aerosol variations on clouds (Yuan and Li, 2010; Yuan et al., 2011).

## 2.5 A new cloud database product

Monthly product files are produced that accumulate all clouds observed during the month. A common set of measurables are provided for each cloud in the product file, which are listed in Table 1 (Wilcox et al. 2023).

Currently the complete dataset consists of all clouds observed using the Detect and Spread algorithm with thresholds applied as described in Sect. 2.1 above for all MODIS granules with at least one sample between 30° N and 30° S latitudes. The data

are screened so that only clouds where the location of the minimum IR brightness temperature within the cloud is within the 30° S to 30° N latitude band. This tropical cloud database is available for all longitudes between these latitude bands for Terra MODIS from 2001 through 2020 and for Aqua MODIS from 2003 through 2020.

The clouds are further screened to identify clouds that are bordering the edge of the MODIS granule. A flag is included in the product files that identifies those clouds that are bordering the edge of the swath and potentially only having a portion of the cloud inside the observation area of the granule. At edges of the scan swath, this is accomplished by recording the maximum viewing zenith angle of the MODIS instrument for each cloud. Clouds with a maximum view zenith angle greater than 64.5 degrees are flagged as clouds at the edge of the granule. Clouds at the along-track edges of the swath are screened based on the index of the 1km infrared samples associated with each cloud. For simplicity in the processing, the along-track indices of each 1 km sample in the MODIS level 2 product are gridded on the same 0.125° lat/lon grid as the retrieval products. By inspection, it is found that clouds that contain a grid box with a MODIS along-track index less than 6 and greater than 2023 reliably flag the clouds that border the along-track edges of the granule. The flag is included as a variable in the product files with a value of 1 for clouds bordering the edge of the granule and 0 for clouds not bordering the edge of the granule.

## 3. Geographic distribution of deep convective clouds

The deep convective clouds from all MODIS-Aqua 5-min. granules during the year 2005 with deep convective clouds observed between 30° S and 30° N latitude is shown in Fig. 3. The location of each cloud corresponds to the 0.125° grid box corresponding to the minimum IR brightness temperature within the boundary of the cloud. The location thus does not correspond to the geographic center of the cloud area but corresponds better to the deepest convective core area within the cloud boundary. The individual cloud locations are aggregated over a 1° lat/lon grid and the total cloud per month in each grid cell is normalized by the size of the grid cell since the grid cells are not equal in area.

Tropical deep convective clouds are most prevalent in the west Pacific Ocean warm pool, the eastern tropical Indian Ocean, the Amazon Basin in South America, and the Congo Basin in Africa. Other prominent features include the Intertropical Convergence Zones (ITCZ) in all equatorial ocean areas, the South Pacific Convergence Zone east of Australia, Southeast Asia, Eastern China, and Central America.

### 3.1 Signature of ENSO variability

During El Niño conditions there is a well-documented warming of the central Pacific Ocean associated with a deepening of the oceanic thermocline and commensurate reduction in the upwelling of cold water beneath the trade winds of the intertropical convergence zone (Bjerknes 1969). This anomalous warming leads to an adjustment of the Walker Circulation such that the deep convection associated with the upward branch of the equatorial overturning circulation in the atmosphere shifts from the western Pacific Ocean to the central Pacific Ocean (Ropelewski, and Halpert 1989; Trenberth et al. 1998). For the initial test

of the deep convective cloud database we chose 2005 and 2011 to explore the contrast in the clouds between an El Niño year
(2005) and a La Niña year (2011). Fig. 4 indicates both the difference in sea surface temperature between the two years and
the expected shift in the distribution of the occurrence of deep convective clouds during the El Niño year compared to the La
Niña year. Fewer deep convective clouds are found over the western Pacific Warm Pool region around Indonesia and northern
Australia in 2005 compared to 2011 while there is a greater number of deep convective clouds in the central Pacific Ocean in
2005 compared to 2011 where the ocean is warmer by greater than 1° C. Other important regional differences in cloud
distribution include decreases in deep convective cloud occurrence in southern Africa, the Atlantic Ocean ITCZ, and portions
of tropical South America. These represented differences between one El Niño year and one La Niña year. A more
comprehensive evaluation of differences in deep convective cloud distributions between El Niño and La Niña conditions is
possible with the greater than 20 years records from Terra and Aqua.

## 4. Size distributions of deep convective clouds

Convective clouds span a broad range of scales from as small at 10s of meters, in the case of small cumulus, to greater than
100,000 km$^2$ in the case of the mesoscale convective systems composed of numerous deep convective cores bound by a broad
stratiform anvil (Houze and Betts 1981). The size distribution of clouds resulting from the application of Detect and Spread to
all MODIS granules between 30 S and 30 N in 2005 is shown in Fig. 5a. The clouds are shown separately for land, ocean, day
and night. That there are fewer clouds over land at all scales is simply a reflection of the fact that there is less land area than
ocean area in the tropical belt. There is a stronger diurnal cycle in convective cloudiness over land than ocean (Hendon and
Woodberry 1993), which is reflected here as a substantially greater number of large clouds during the daytime overpass at
1300 local time than the night-time overpass at 0100 local time, and a substantially greater number of small clouds at 0100
local time over land compared to 1300 local time. By 1300 local time the solar heating of tropical land surfaces has contributed
to enhanced daytime CAPE, which in-turn increases the deep convection (Begman and Salby 1996; Dai 2001).

The power-law nature of the cloud size distribution has been well documented and known to span several orders of magnitude
(Houze and Cheng 1977; López 1978). Garrett et al. (2018) argue that the total amount of convective cloudiness in a region is
controlled by the total static stability of the region and that the robust scale-invariant nature of the cloud size distribution arises
from competition among the circulations at the edges of the clouds for this available energy.

The size distributions of the convective objects in the population shown here departs somewhat from the expected power-law
size distribution. This is partially a consequence of pixelated nature of clouds that are only one to a few pixels in size. These
samples may either be overcast pixels at or colder than the threshold temperature, or perhaps partially filled pixels with cloud
that are colder than the threshold temperature. To evaluate the effect of resolution on the size distribution of clouds in the
database we compare the database of clouds resulting from averaging the 1 km MODIS brightness temperatures on the 0.125°
grid as described in Sect. 2.2 above to a similar set of clouds resulting from averaging the same brightness temperatures to a
finer 0.025° grid (Fig. 5b). Clouds at the scale of only one or a few grid cells are severely under-sampled compared to clouds

sampled from data at a substantially higher resolution. For clouds larger than about 4000 km$^2$, which is equivalent to a cloud of about 5 grid cells on a side, the error in the number of clouds a particular size can be as large as a factor of two. This error is an undercount at the small end of that range and an overcount of very large clouds.

At the largest scales, clouds approach $10^6$ km$^2$ in size. To evaluate whether the swath width of MODIS clouds is sufficient to
capture the shape of the size distribution of clouds at these larger scales we compare the results from MODIS over the oceanic portions of the northern Indian Ocean to the size distributions in the Roca and Ramanathan (2000), which used the identical Detect and Spread methodology and thresholds, but applied it to INSAT1B geostationary satellite data, which is not subject to the limited swath width of MODIS (Fig. 5c). Since the sample sizes are different, the curves have been normalized by the total number of clouds observed in each population. This comparison shows that MODIS has sufficient swath width to capture even
the largest clouds using the Detect and Spread thresholds chosen for this database and that the clouds at the largest scales are not under-sampled by MODIS as would be expected if the swath width were too narrow.

### 4.1 Structure of deep convective clouds

The scales of cloud systems are strongly related to the structural evolution of convective clouds. For clouds that grow to become mesoscale convective systems, the development of a broad region of stratiform cloud contributes to a substantial
growth in the scale of cloud systems (Roca and Ramanathan 2000). Thus, there is a close connection between the level of mesoscale organization of cloud systems and total thermodynamic forcing of these cloud systems from radiative forcing and latent heating (Wilcox and Ramanathan 2001; Nesbitt et al. 2006; Feng et al. 2019). The distribution of infrared brightness temperatures within the cloud can give some rough clues to elements of the structure of the cloud systems. For example, Fig. 6a shows that there is a systematic relationship between the minimum infrared brightness temperature and the horizontal scale
of cloud systems. The larger clouds clearly have deeper convective cores (Roca and Ramanathan 2000; Cetrone and Houze 2006). The tropics-wide data indicate that the relationship is quite robust. There is some difference between the daytime and night-time relationship between minimum infrared brightness temperature and cloud scale, which is more evident over land than ocean and likely reflects the stronger diurnal variation of convection over tropical land surfaces than ocean surfaces with convection more frequently initiated during daytime and maturing in the night-time such that the same depth of convective
core is more frequently associated with a larger, more mature cloud shield at night than during the day.

The minimum 89 GHz PCT does not show a strong dependence on cloud scale (not shown), but rather the average of the 89 GHz PCT over the area of the cloud does (Fig. 6b). These results imply that the deepest layer of vertically integrated ice may be relatively similar across clouds of various scales, which is a result that requires further inquiry, but that area of the cloud with significant scattering from precipitation-sized ice not only increases with cloud scale and represents an increasing share
of the total cloud system area as the cloud scales increase. This result is consistent with the results from the Northern Indian Ocean convection reported in Roca and Ramanathan (2000) who examine the fraction of cloud area colder than 220 K in mature convective clouds and find that it is less for moderately sized clouds compared to the smallest clouds, but increases with cloud size from the moderate to the large cloud systems. They argue that this is a consequence of the mesoscale

organization of the larger cloud systems. Based on the results of the Roca and Ramanathan (2000) study, we have included
similar metrics for the fraction of cloud colder than specific temperature thresholds so that the regional variation of such consequences of mesoscale organization can be explored, as well as how the size and structural metric relate (or do not relate as found by Igel and van den Heever 2015) to environmental characteristics of the atmosphere as discussed further in the next section.

Note that the results shown here are essentially averages over snapshots of cloud systems at varying stages of their lifecycle.
Further insights into the structural evolution of deep convection as indicated by these metrics can be achieved by linking the clouds observed in this database to cloud systems tracked in geostationary satellite data, as discussed further in section 5 below. For example, the relationship between the ratio of the cold cloudiness to overall cloud system size changes as mesoscale convective systems evolve (e.g. Wilcox 2003; Elsaesser et al. 2022).

**4.2 The role of convective available potential energy and vertical shear of the horizontal wind**

The tropical atmosphere is in a general state of conditional instability where the magnitude of the instability is determined by the vertical profiles of temperature and humidity. Greater levels of instability in the column are naturally expected to produce stronger convection and larger cloud systems (Zawadski et al. 1981; Adams and Souza 2009). In addition to the level of instability, the vertical shear of the horizontal wind is also critical to the overall structure and organization of convective cloud systems (Moncrieff 1992). Greater shear can adjust the convective outflow in the anvil such that new convective elements
within a cloud cluster can penetrate higher without interacting with other convective elements, leading to larger and longer-lived cloud systems (Rotunno et al. 1988). Though the CAPE and shear impact the strength of convection and the associated size and structure of the resulting cloud systems, these quantities are certainly not predictive of cloud scale. As shown below significant regional variations in the relationships between the environmental conditions and size and structure of clouds systems are apparent. To better explore these relationships in large statistical ensembles of clouds, we have collocated the
MERRA-2 profiles of temperature, humidity and winds so that these quantities can be evaluated for each cloud system.

The joint dependence of convective cloud system size on CAPE and shear is shown for the global tropics for all cloud systems observed between 30S and 30N during 2005 in Fig. 7. Here, the CAPE and shear values shown in the figure are the maximum value of the CAPE and shear within the boundary of the cloud, and the shear is defined as the change in relative wind speed between the 875 hPa level and the 250 hPa level. In general, the cloud size increases systematically with both CAPE and shear
from hundreds of square kilometers at low CAPE or shear to greater than 50,000 km$^2$ for clouds forming in a high CAPE and shear environment. Though the increase in cloud size with CAPE and shear appears robust and systematic in these data, there is some spatio-temporal variability in the relationship. For example, over land, where the diurnal cycle of deep convection is most significant, cloud systems at night are larger than during the day for the same values of CAPE and shear. There is a stronger relationship between the scale of cloud systems and maximum value of CAPE and shear within the cloud boundary
than for the average value of CAPE and shear within the cloud boundary, or the values of CAPE and shear at the location of

the minimum infrared brightness temperature within the cloud (not shown). This likely reflects that there is not a perfect correspondence in space between convection observed by MODIS and simulated convection in MERRA-2.

To further illustrate the geographic differences we compare winter monsoon oceanic clouds over the tropical Indian Ocean (Fig. 8a,b) with southern hemisphere summer continental clouds over the Amazon (Fig. 8c,d). The CAPE and shear environments are significantly different in the two regions. The range of CAPE values over the Amazon extends to significantly higher values than over the tropical Indian Ocean. However, the range of shear values over the tropical Indian Ocean extends to higher values than over the Amazon. At intermediate values of CAPE and shear, clouds over the tropical Indian Ocean are generally larger than those over the Amazon, suggesting that regions with more extreme values of CAPE and shear do not necessarily exhibit larger cloud systems. Thus, while cloud system size may not be predictable from CAPE and shear values, the systematic dependence of cloud size on these parameters, in addition to their geographic variations, can provide useful constraints on the representation of convection and clouds in atmospheric models. Furthermore, the exploration of the evolution of these relationships over the lifecycle of cloud systems (as discussed further below) may provide additional insights into the roles of CAPE and shear on cloud scale evolution.

## 5. Summary and future applications of the cloud database

Here we have presented a new cloud database that is being made publicly available on the NASA Goddard DAAC. The database contains 10s of millions of deep convective clouds that includes the size of the cloud, associated structural and microphysical properties of the clouds, and aspects of the environment that the clouds reside within. This database has been designed with several specific applications in mind, which will be the subject of follow-on studies.

Toward the goal of providing an empirical dataset against which atmospheric models may be assessed, this dataset has been constructed with the idea that similar cloud databases can be constructed from model output. While comparisons are possible even with coarse grid global climate models (e.g. Wilcox 2003), the advent of large-domain, and even global-scale, cloud resolving models allow for the possibility for comparisons between satellite-derived cloud statistics and model simulated clouds at a variety of scales, as demonstrated in a handful of other studies (Skok et al. 2010; Caine et al. 2013; Heikenfeld et al. 2019). Of particular interest is the application of the database constructed here to determine if the simulated relationships of cloud scales and structure to CAPE and shear are similar to the observed relationship, and if the regional variations can be captured. Likewise, the dataset can be used to evaluate whether simulated structural and microphysical properties for clouds across scales are consistent with the statistics found in this database. Though CAPE and shear are not predictive of cloud scales and structure, the differences between the observed relationships and model-simulated relationships are expected to indicate pathologies in the parameterization of cumulus convection and convective cloudiness, particularly in models with unresolved convection where the convective closure may be parameterized based on CAPE. This can be achieved using statistics derived from very large ensembles of observed clouds, including robust statistics across regional variations and interannual climate variations. Given that the exact numbers of clouds observed at a given scale varies with the resolution of data used to construct

the cloud database, as seen in figure 5 above, some further work is required to determine how the resolution of the underlying data relates to the grid resolution of the model being evaluated, how the empirical relationships evident in this database may
depend on the resolution of the underlying data used to identify the clouds, and how that uncertainty compares with the differences between the model and the observed clouds.

Several hypotheses have been proposed for mechanisms by which aerosol may alter the structure and size of deep convective clouds (e.g. Koren et al. 2010). Such changes in the coverage of these clouds may contribute important impacts on the radiative forcing of cloud systems beyond those that might be attributed just to changes in cloud microphysics. Furthermore, the
environmental factors, including CAPE and shear may have an impact on the response of cloud systems to aerosol perturbations. For example, the invigoration of convection attributable to microphysical modifications of convective clouds by aerosols (Koren et al. 2005) may be more effective in weakly sheared environments compared to strongly sheared environments (Fan et al. 2009). The dataset described here was constructed in part to test these ideas by capturing not just estimates of the aerosol loading in the vicinity of clouds, but also the thermodynamic properties of CAPE and shear that
strongly influence the scale of cloud system. By controlling for these, it is possible to compare a large sample of clouds forming in a similar thermodynamic environment, but differing aerosol amounts.

Data from MODIS and AMSR-E, which offer sampling only from a polar orbit, were chosen for the long, greater than 20-year record, and the ability to provide retrievals of quantities such as the cloud droplet effective radius and the 89 GHz polarized corrected temperature, which provide clues to microphysical aspects of the cloud that we anticipate have important links to the
size and structure of deep convective cloud systems. A limitation of this database compared to other recent attempts to track clouds through their lifetime is that polar orbiting satellites do not provide the temporal sampling to allow cloud tracking. Thus, in our analysis we are mixing clouds that may be early in their development and growing with clouds that may be late in their lifecycle and decaying. While there are important questions related to the impacts of atmospheric thermodynamics and aerosols on the lifecycle of deep convective clouds that must be addressed using cloud tracking algorithms applied to
geostationary data, we believe that the insights available from the database described here are complementary to the tracking results. Furthermore, it is likely that geostationary cloud tracking datasets, such as described in Fiolleau and Roca (2013), because of the similarity in the methodology applied, can be used to place clouds in the dataset presented here within the context of their lifecycle.

**Code availability**

The codebase for producing the cloud database and the figures contained in this manuscript are publicly available at https://github.com/emwilcox-dri/NASA-MEaSUREs-deep-clouds.

**Data availability**

The datasets described in this paper may be obtained from the NASA Goddard Earth Sciences Data and Information Services Center (GESDISC) at the following DOIs: 10.5067/0Y3M8IO4XKG5 for MODIS/Aqua and 10.5067/HXRNU4HCIPA6 for MODIS/Terra.

**Author contribution**

EW and TY designed the study and the deep cloud database; EW and HS wrote code to produce the cloud database and the product files; EW analyzed the data; EW wrote the manuscript draft; TY and HS reviewed and edited the manuscript.

**Competing interests**

The contact author has declared that none of the authors has any competing interests.

**Acknowledgements**

This work has been supported by NASA grants NNX11AG89G and 80NSSC18M0084.

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

**Table 1: Variables and their dimension in monthly data product files. Number of clouds is the total number of cloud identified by the Detect and Spread algorithm for the entire global tropics from 30° S to 30° N.**

| Variable | Dimension |
|---|---|
| MODIS aerosol optical thickness averaged on 5° lat/lon | Number of clouds |
| MERRA-2 aerosol optical thickness at native resolution at location of minimum IR brightness temperature | Number of clouds |

| | |
|---|---|
| MERRA-2 aerosol optical thickness averaged over cloud area | Number of clouds |
| MERRA-2 convective available potential energy at native resolution at location of minimum AMSR-E 89 Ghz polarized corrected temperature | Number of clouds |
| MERRA-2 convective available potential energy at native resolution at location of minimum infrared brightness temperature | Number of clouds |
| MERRA-2 convective available potential energy at averaged over cloud area | Number of clouds |
| Change over 6 hours of MERRA-2 convective available potential energy at averaged over cloud area | Number of clouds |
| Area of cloud | Number of clouds |
| Cloud classification | Number of clouds |
| Fraction of cloud area with infrared brightness temperature colder than 200 K | Number of clouds |
| Fraction of cloud area with infrared brightness temperature colder than 210 K | Number of clouds |
| Fraction of cloud area with infrared brightness temperature colder than 220 K | Number of clouds |
| Fraction of cloud area over land | Number of clouds |
| Infrared brightness temperature at the location of minimum AMSR-E 89 GHz polarized corrected temperature | Number of clouds |
| latitude at location of minimum AMSR-E 89 GHz polarized corrected temperature | Number of clouds |
| latitude at location of minimum infrared brightness temperature | Number of clouds |
| longitude at location of minimum AMSR-E 89 GHz polarized corrected temperature | Number of clouds |
| longitude at location of minimum infrared brightness temperature | Number of clouds |
| Maximum value of MERRA-2 convective available potential energy within cloud area | Number of clouds |
| Maximum change over 6 hours of MERRA-2 convective available potential energy within cloud area | Number of clouds |
| Maximum MERRA-2 vertical shear of horizontal wind within cloud area | Number of clouds |
| Maximum MODIS viewing zenith angle within cloud area | Number of clouds |
| Minimum value of 89 Ghz polarized corrected temperature within cloud area | Number of clouds |
| Minimum value of infrared brightness temperature within cloud area | Number of clouds |
| Minutes since January 1 1990 0000 GMT | Number of clouds |
| MODIS particle effective radius averaged in bins of infrared brightness temperature | Number of clouds x number of infrared bins |
| AMSR-E 89 GHz polarized corrected temperature at location of minimum infrared brightness temperature | Number of clouds |
| AMSR-E 89 GHz polarized corrected temperature averaged over cloud area | Number of clouds |
| MERRA-2 vertical shear of horizontal wind at native resolution at location of minimum AMSR-E 89 GHz polarized corrected temperature | Number of clouds |
| MERRA-2 vertical shear of horizontal wind at native resolution at location of minimum infrared brightness temperature | Number of clouds |
| MERRA-2 vertical shear of horizontal wind at native resolution averaged over cloud area | Number of clouds |

| | |
|---|---|
| MERRA-2 wind direction at the 200 hPa pressure level at the location of minimum infrared brightness temperature | Number of clouds |
| MERRA-2 wind direction at the 500 hPa pressure level at the location of minimum infrared brightness temperature | Number of clouds |
| MERRA-2 wind direction at the 300 hPa pressure level at the location of minimum infrared brightness temperature | Number of clouds |
| Flag for cloud bordering granule edge | Number of clouds |


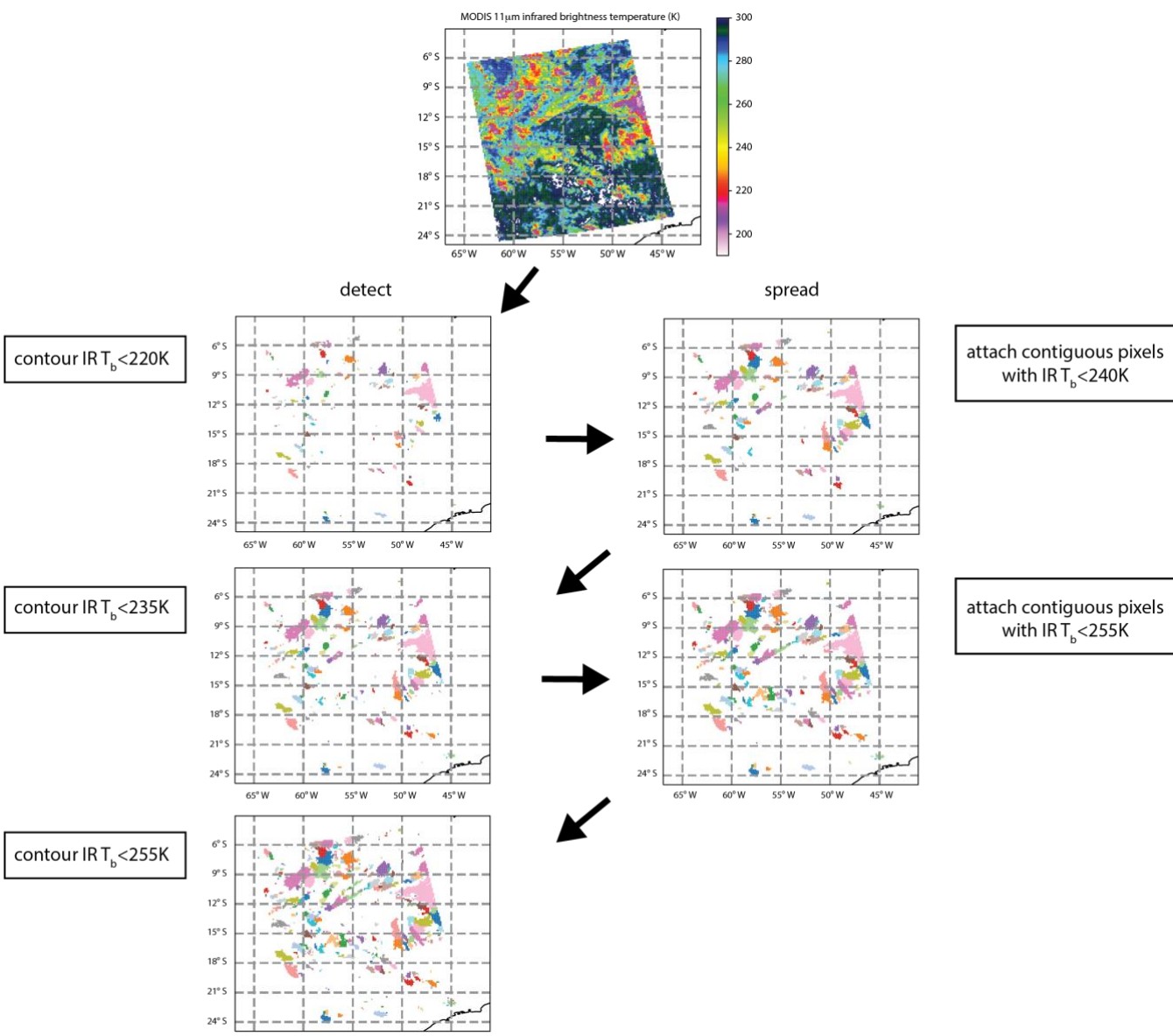

**Figure 1: Example of the progression of the Detect and Spread algorithm as applied to each 5-minute granule of MODIS infrared brightness temperature ($T_b$) gridded on a 0.125° lat./lon. grid. Top panel shows the original MODIS infrared image. Rows below show the consequence for detecting cloud objects of subsequent Detect and Spread steps. Colored areas show individual cloud objects. Colors are assigned randomly. Example is from Aqua MODIS on 27 January 2011, 17:20 UTC located over South America.**

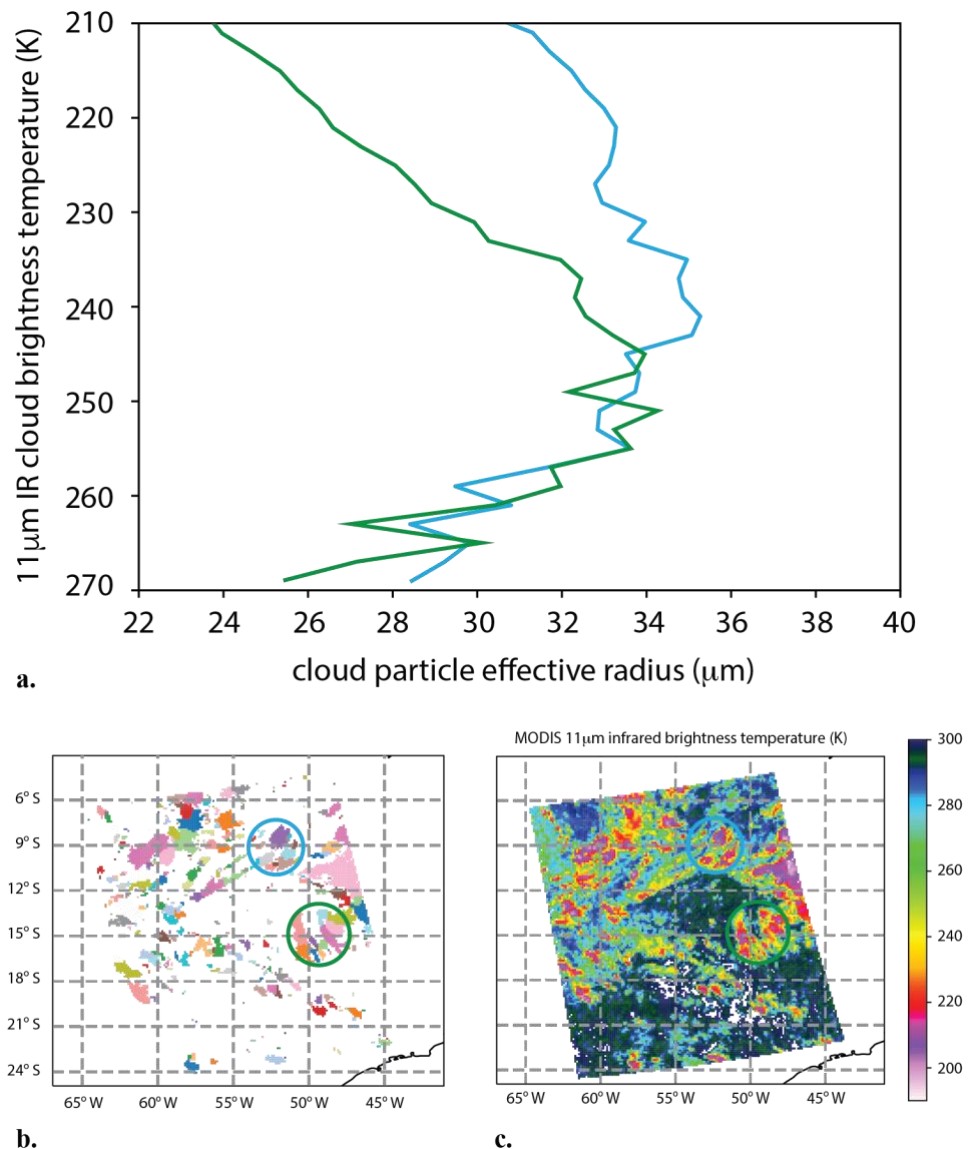

**Figure 2: (a) Average of MODIS-retrieved cloud hydrometeor particle effective radius against the collocated MODIS infrared cloud top temperature for two clusters of approximately 25 convective clouds each in a granule of Aqua MODIS on 27 January 2011, 17:20 UTC located over South America. (b) Map of the convective clouds identified by the Detect and Spread algorithm where each randomly colored object represents and individual cloud object. Colored circles indicate the cloud objects averaged in the composite profiles of the same color in (a). (c) The infrared cloud image from the corresponding granule used to identify the clouds and construct the composite profiles of particle effective radius.**


## 2005

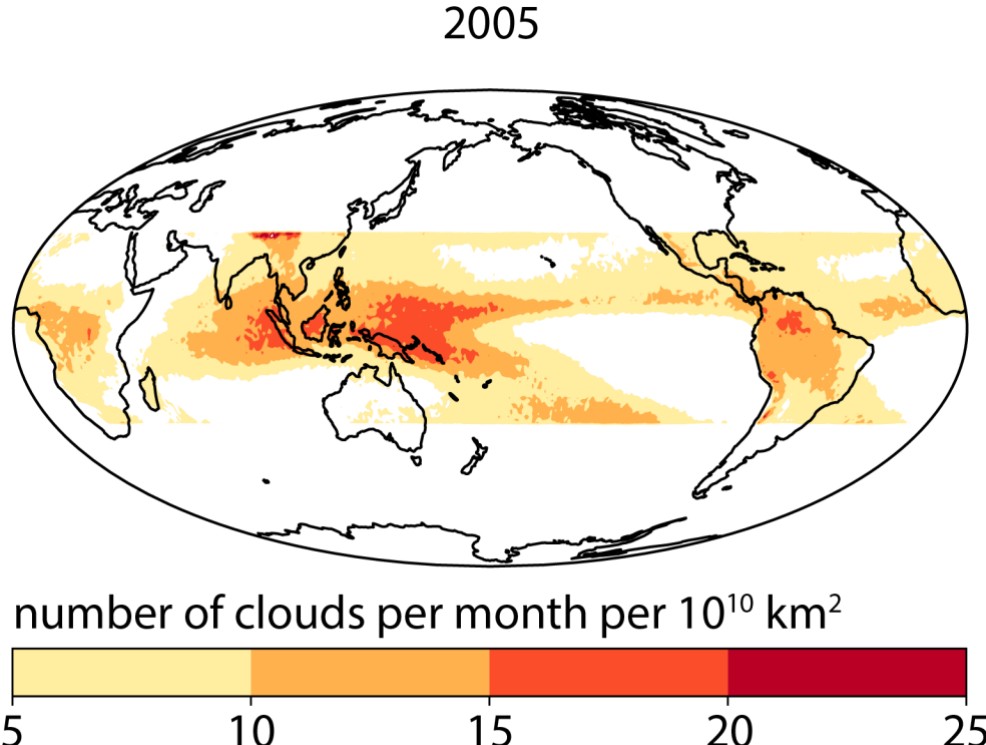

number of clouds per month per $10^{10}$ km$^2$

**Figure 3: The number of deep convective clouds per month per $10^{10}$ km$^2$ for all tropical 5-minute MODIS granules in the year 2005 in the global tropics between 30° S and 30° N.**

## 2005 minus 2011

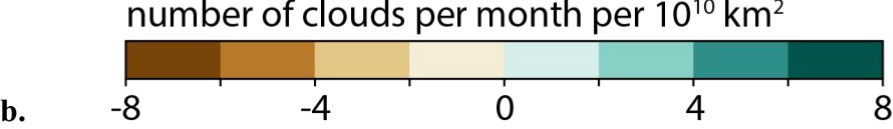

**Sea surface temperature difference (deg. C)**

-2          -1          0          1          2

**a.**

**number of clouds per month per $10^{10}$ km²**

-8          -4          0          4          8

**b.**

**Figure 4: (a) Difference between the 2005 and 2011 annual averaged sea surface temperature from the version 2 of the NOAA Optimal Interpolation dataset (Reynolds et al. 2002). (b) Difference between the 2005 and 2011 annual averaged number deep convective clouds per month per $10^{10}$ km² in the global tropics from 30° S to 30° N.**


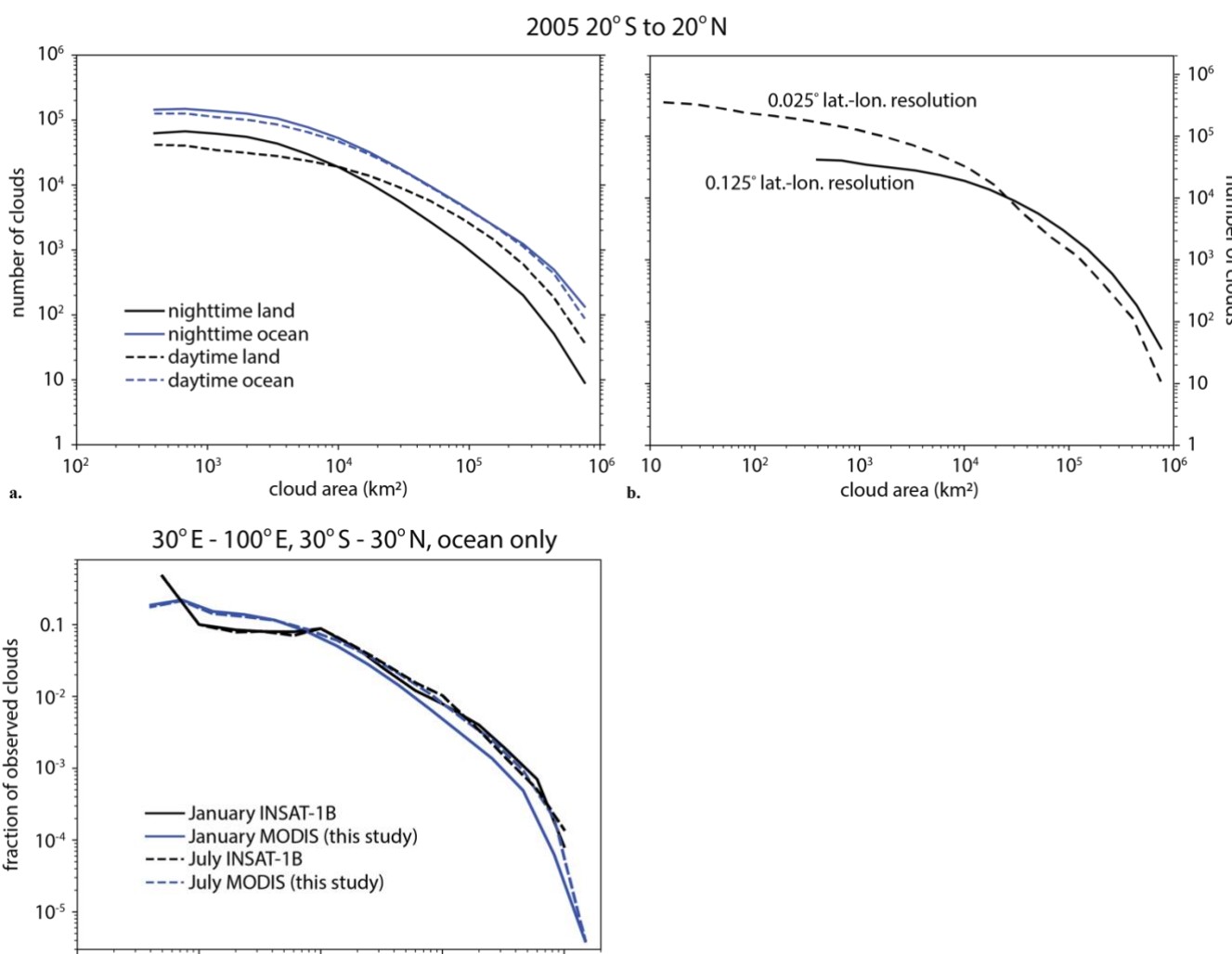

**Figure 5: (a) The number distribution of deep convective clouds for the global tropics from 20° S to 20° N. Black lines are clouds over land and blue lines are clouds over ocean. Solid lines are clouds observed at night, dashed lines are clouds observed during day. (b) The number distribution of all deep convective clouds in the global tropics from 20°S to 20°N where the solid line is the application of the Detect and Spread algorithm to 1 km MODIS infrared brightness temperature gridded on a 0.125° lat./lon. grid and the dashed line is the application of Detect and Spread to the same data gridded on a 0.025° lat./lon. grid. (c) The number distribution of deep convective clouds normalized by the total number of observed clouds over the tropical Indian Ocean where black lines are for all MODIS granules in the years 2005 to 2011 and the blue lines are from INSAT-1B geostationary satellite from the year 1999 (adapted from the sum of the curves in Fig. 3 of Roca and Ramanathan [2000]). Solid lines are January and dashed lines are July.**

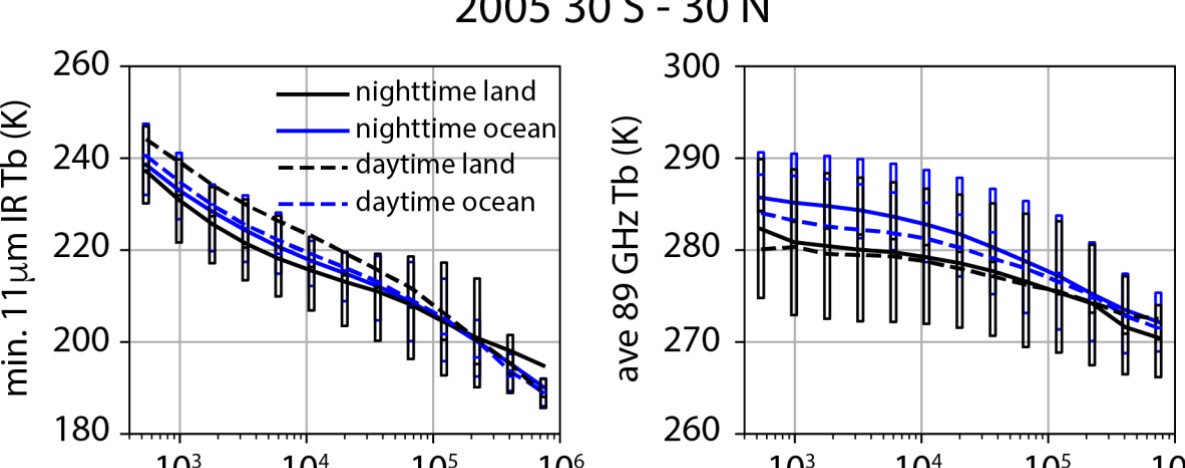

**Figure 6: (a)** Minimum value of infrared brightness temperature within the deep convective cloud against area of the cloud. **(b)** Average over the cloud area of the 89 GHz polarized corrected brightness temperature against the area of the cloud. Black lines are clouds observed over land and blue lines are clouds observed over ocean. Solid lines are clouds observed at night and dashed lines are clouds observed during day. Data are from all overpasses of the Aqua satellite during the year 2005 between 30° S and 30° N. Boxes indicate the range of values from the 25th to 75th percentile of the range of values in each cloud area bin and correspond to the solid lines (nighttime clouds).

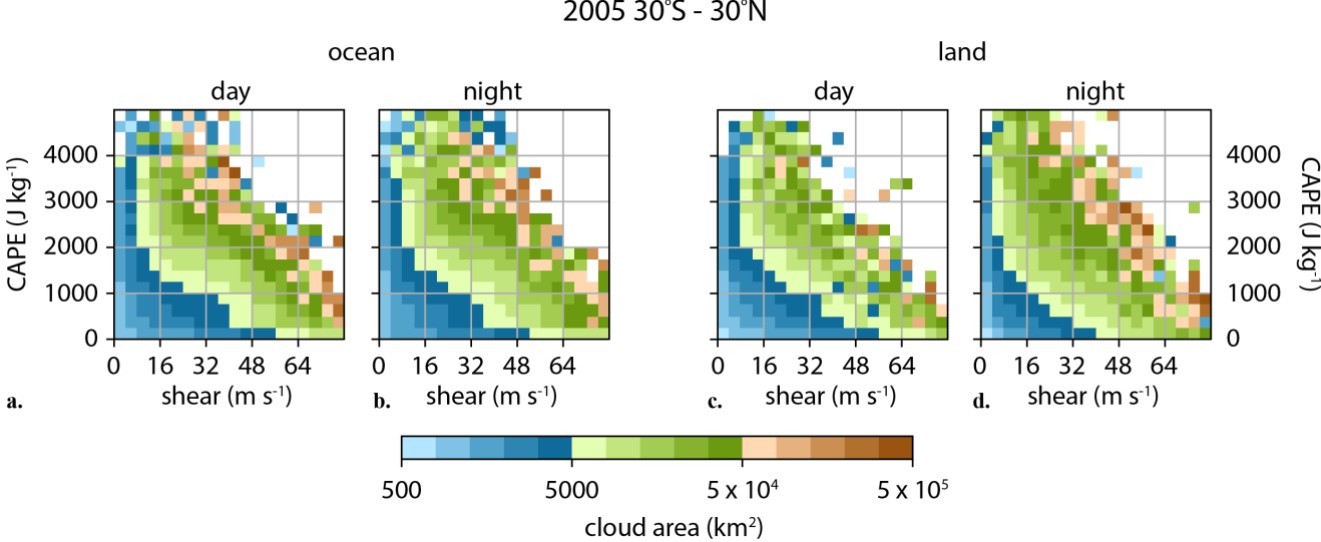

**Figure 7: The average cloud area in bins of maximum CAPE and vertical shear of the horizontal wind for deep convective clouds observed from all overpasses of the Aqua satellite during the year 2005 between 30° S and 30° N. The total number of clouds observed is N= 4,062,508. (a) Oceanic clouds observed during day, (b) oceanic clouds observed at night, (c) land clouds observed during day, (d) and land clouds observed at night.**

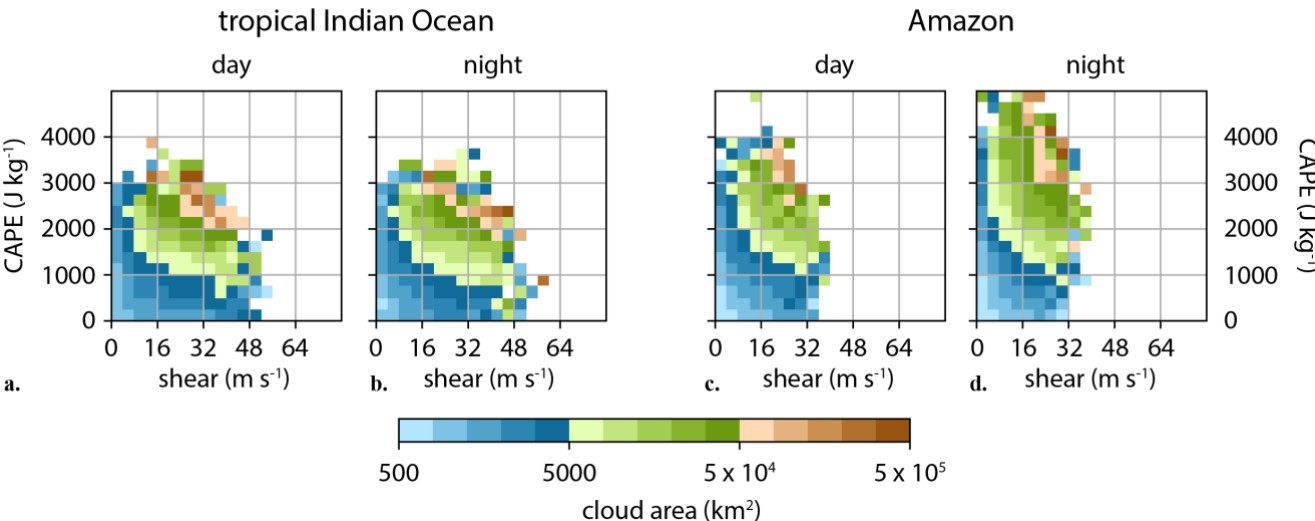

**Figure 8: The average of cloud area in bins of maximum CAPE and vertical shear of the horizontal wind for all deep convective**
**clouds observed over (a) and (b) the tropical Indian Ocean (10° S – 15° N, 50° E – 100° E; number of clouds N = 910,549), and (c)**
**and (d) the Amazon (10° S – equator, 50° W – 65° W; number of clouds N = 205,757). For both locations data include deep convective**
**clouds observed from all overpasses of the Aqua satellite during the months Nov.-Mar. from the eight years 2004-2011.**