# Peer review of "Deep convective cloud system size and structure across the global tropics and subtropics"

_Atmospheric Measurement Techniques, 2023_

## Author Comment (AC1)

Response to reviewer 1

*Major Comments:*

*System flagging (#1) – what do the authors do for systems whose boundaries reach the MODIS swath edge? Are those systems flagged? Such systems are always going to be under-reported in size, since some unknown (greater or equal to 0) fraction extends beyond the viewing area, which therefore will bias future science analyses. If MODIS pixels touch the edges of the swath granule (in the swath or in the pixel direction), I think inclusion of an "edge" flag (binary: e.g., 0 for not touching, 1 for touching) would be useful. That way, users who would like to use this database can decide their own comfort level for using systems fully within the viewing area for statistical analysis versus not using systems if they are not confident that their size is accurately reported. By including a flag as opposed to removing, the developers are not forcing users to make a decision.*

This has been implemented. The updated product files will include this flag when they are publicly posted by NASA. The text describing the procedure for identifying cloud bordering the edge of the granule has been added to the manuscript in the paragraph beginning page 8, line 249 in the tracked-changes version of the revised manuscript and in Table 1. For the across-track edges, the clouds bordering the granule edge are identified by tracking the maximum view zenith angle of each cloud. The view zenith angle is highest at the two edges of the swath. Screening clouds at the along-track edges of the granule is slightly less straightforward because of the irregular spacing of the 1 km samples in the MODIS level 2 files. Here, we simply grid the along-track index of the 1 km IR brightness temperature samples in the two-dimensional (along-track and across-track) ordering of the samples in the level 2 files and determined minimum and maximum thresholds on that gridded along-track index that reliably screen clouds that border the along-track edge of the swath.

*System flagging (#2) – regarding use of MERRA-2 outputs (CAPE, etc.) mapped to the identified systems. Have the authors plotted latitude-longitude maps of MERRA-2 convective systems (e.g., map of OLR or rainfall rates?) alongside maps of identified convective systems from MODIS? For any day & time, they often do not resemble each other (unless it is an O(1000km) system or mid-latitude system). MERRA-2, although it assimilates observational data, more often than not, does not simulate individual systems at the same time/location (unsurprising since MERRA-2 convective systems are dependent on their convective parameterization). Many times, MERRA-2 produces a convective system where the observations do not indicate one exists (or vice versa). For mesoscale environments quickly modified by convection (particularly diagnostics influenced by the planetary boundary layer [PBL] characteristics), this issue might impact CAPE computations (or anything related to T and Qv) since not having a convective system in MERRA-2 leads to the PBL remaining "undisturbed" and characterized by a buoyancy metric that differs from reality. I think a) it should first be determined if an equivalent convective system was identified in MERRA-2 via some determination of whether rainfall was beyond some threshold over the MODIS convective system area and/or OLR was below some threshold for the same time/space locations as the observed MODIS system, and b) a MERRA-2 flag should then be*

*derived such that if a convective system is not found in MERRA-2 at the same time/place: e.g., 0 is reported for no equivalent system existing in the reanalysis; and 1 if MERRA-2 itself is simulating a system going on at the same time/place an observed system is evolving. Having this flag allows users to have confidence in MERRA-2 environments (or to use their own filtering) if it can be known in advance that the MERRA-2 environment is at least approximately resembling observed mesoscale convective environments.*

Before addressing the reviewer's suggestion for a flag, we note that one way in which the results shown in figures 7 and 8 may partly account for the possible mismatch in the location of deep convection between MERRA-2 and the observations, is that the values for CAPE and shear shown in the figures are the maximum values of CAPE and shear within the cloud boundary. Thus, for larger clouds where the observed distribution of infrared brightness temperatures may not precisely correspond to the location of convective cloudiness in MERRA-2, the analysis may still capture local maxima in CAPE and shear if they are within the boundary of the cloud. This point was and remains included in the text in section 4.2. This has been further clarified in the captions for figures 7 and 8 in the revised manuscript. This choice was made after experiments with using the average CAPE and shear values within the cloud boundary, as well as the values at the location of the minimum in MODIS infrared brightness temperature. The use of the maximum values yields the clearest and most systematic relationship between CAPE, shear and cloud system size. The point made by the reviewer about the spatial correspondence between MERRA-2 and the observed convection may be an important reason for this. This point has been further clarified in the paper (p.11 lines 378-384 in the tracked changes version of the revised manuscript).

We have followed up on the reviewer's suggestion to explore co-located maps of the MODIS infrared brightness and MERRA-2 outgoing longwave radiation. Some examples from a variety of scenes containing observed and simulated convection are included in a new set of supplementary figures (Supplementary Fig. 1). Indeed, as the reviewer notes, while in many of the scenes there is a correspondence between the observed and simulated convection, they do not indicate a perfect 1:1 correspondence. Nevertheless, we struggled with how to capture this in a simple flag as suggested by the reviewer. We reflect a little more on the challenges in the following discussion, and then discuss how the manuscript has been revised to try to address the reviewer's concern and clearly convey to the reader the lingering uncertainties in matching MODIS observations to MERRA-2 reanalysis products.

A fundamental challenge in applying a flag to the data, as suggested by the reviewer, is that metrics of convective intensity, such as precipitation or outgoing longwave radiation are continuous variables. Likewise, the level of correspondence between convection observed in satellites and simulated in the MERRA-2 reanalysis will be similarly continuous. In contrast, a simple flag as suggested by the reviewer is by definition binary and requires picking a threshold. Picking a threshold that is not itself misleading is a particular challenge.

After examining the MERRA-2 simulated precipitation we decided not to try to use precipitation for the purpose suggested by the reviewer. The highly nonlinear physics of convective

precipitation production, and its relatively simplistic representation in course resolution global models, adds considerable additional uncertainty to the goal of determining a reasonable threshold for a flag.

We examined direct scatter plots of the 0.125 degree latitude/longitude gridded MODIS infrared brightness temperatures with the nearest neighbor matched MERRA-2 outgoing longwave radiation at the native MERRA-2 resolution. The resulting clouds of points were inconclusive in terms of determining which clouds might correspond better to simulated convection than others (not shown).

Potentially more useful is an examination of scatter plots of the minimum MODIS infrared brightness temperature with the minimum MERRA-2 outgoing longwave radiation for each cloud. The resulting scatter plot for all of the clouds in the scenes included in the supplementary images is included as Supplementary Fig. 2. The scatter is shown for all clouds, as well as clouds of varying minimum cloud size. This is done to examine the notion that the level of correspondence between these observed and simulated values is likely to improve with cloud size. It shows that there is a rough correlation between these quantities, although with considerable scatter. The minimum infrared brightness temperature is concentrated at lower values as the minimum cloud size threshold is increased (i.e. progressing the Supp. Fig. 2 panel a to panel d). This occurs because the minimum infrared brightness temperature systematically decreases with increasing cloud size as shown in Fig. 6a.

For relatively warm values of minimum infrared brightness temperature, which are predominantly the smaller clouds, there is essentially a continuous range of OLR values in the nearest-neighbor MERRA-2. Likewise, there is a hint of a potential threshold of 220 W m-2 in outgoing longwave radiation above which MODIS-observed clouds may correspond to locations where MERRA-2 has not generated significant convection.

While there appear to be some outliers where minimum IR brightness temperature is quite cold (<220 K), but MERRA-2 minimum outgoing longwave radiation is relative high (>200 – 220 W m-2). These outliers suggest that these may be cases where MODIS is observing a very cold deep convective cloud top, but the OLR is not indicating intense convection in MERRA-2. One limitation of this as a potential threshold for flagging, however, is that this feature only really stands out for a handful of the very cold minimum IR brightness temperature clouds, which are nearly all very large mesoscale cloud systems. For the small clouds with relative warm minimum IR brightness temperatures (>220 K), the scatter plot is quite a bit more ambiguous and shows basically a mass of point across the range of minimum OLR values, with no clear indication which may be indicating a cases where MODIS deep convection is not corresponding to convection in MERRA-2.

From this, we conclude that a threshold suitable for flagging poor matches may be possible, but also likely will vary depending on cloud scale and/or minimum IR brightness temperature of the cloud. And that any such threshold for flagging would likely become more uncertain or ambiguous as the clouds scale decreases and minimum IR brightness temperature increases. At

present, the MERRA-2 OLR is not a parameter that is captured in the image processing code that generates the products. However, based on this examination, we will be creating an updated version of the production code that will capture the OLR and will seek to include the minimum and average OLR for each cloud in a subsequent version of the product files from which we can pursue a more systematic examination and toward a goal of determining if a reliable flag can be designed to better guide users as suggested by the reviewer.

Discussion that summarizes all of the above and plans for further examination of this issue has been added in section 2.3 (p. 6 lines 167-197 in the tracked changes version of the revised manuscript).

*Minor Comments:*

*Lines 48-50: The authors use MODIS (sun-synchronous orbiter), and I do not think systems are tracked (as can be done with GOES) in this database; writing the word "Lagrangian" is confusing, unless I am missing something, in which case clarification is needed.*
*I am aware of the literature suggesting how cloud top distributions for effective radius (Re) approximate a Re(z) height profile into a cloud; however, this view is not unanimously agreed upon for all convection environments globally in the tropics. Aircraft are the only source of "validation", and not only are these data sparse, aircraft usually only infer sizes adjacent to convection and not inside convection (due to flight restrictions in general, with a few exceptions). It is unnecessary speculation to call cloud top distributions a profile, particularly in convection of varying life stages. Therefore, I recommend relabeling the "profiles" in section 2.4 and figure being shown as "cloud top distributions." Then, the discussion text noting how this may be equivalent to cloud top PDFs can remain for interpretation purposes. From a perspective of introducing an observational database and its utility, it is not clear to me why this assumption has to be pre-supposed for discussing the database and plotting a preliminary result. Spatial variations as a function of cloud top heights themselves are very useful too, independent of the assumption that they represent profiles.*

The use of Lagrangian in this context is misleading. The instance of that word has been removed. The point about the uncertainties in interpreting the distribution of effective radius against brightness temperature as a profile is taken.

We have also followed the reviewer's recommendation and avoided referring to the distributions of cloud drop effective radius against infrared brightness temperatures as "profiles". It is worth noting that the product header does not refer to these data as profiles, even as the manuscript had adopted that interpretation. The title of section 2.4 has been changed to "Cloud-top distribution of droplet effective radius". The discussion of the interpretation of these distributions as estimates of a composite profile of the droplet effective radius in section 2.4 remains, including discussion of the papers that have argued for that interpretation. The word "profile" has also been removed from the caption of figure 2.

*Regarding Fig. 5b, the sensitivity to averaging – to my eyes, those differences are comparable to day/night differences over land in Fig. 5a (or in other words, they appear large). Eyeballing, I see a factor of 5 at the low end, and 2 at the high end. Thus, I do not understand the comment on lines 242 about them being similar with slight over-sampling of large clouds. What about the small end too? Since the authors intend for this dataset to be used as a model evaluation benchmark, and in light of this dependence on averaging, how do the authors propose to apples-to-apples facilitate this database to be used for comparing to a range of models, since averaging does have this impact? Can averaging impacts be incorporated into an uncertainty estimate somehow?*

The discussion in this section regarding similarity of the curves was primarily considering the shape of the curves at the larger sizes and the curvature implying a break from the size distribution scaling of the smaller clouds from $10^4$ to $10^6$ km$^2$. But the discussion was not very clear here. This section has been revised. The magnitude of the differences between the curves are now discussed quantitatively. The revised text appears on page 9, lines 307-310 of the tracked-changes version of the revised manuscript.

The matter of the significance of the resolution dependence for the applications of the dataset is important, if not yet completely reconciled. Obviously, down at the scale of clouds composed of 1 to 10 samples, the size distribution is going to significantly undercount clouds relative to the same algorithm applied to higher resolution imagery. Certainly, at the scales of larger clouds a difference in the number of roughly a factor of two, or comparable to the day-night difference, is important for the size distribution. For a comparison with model simulated clouds, then knowing that the uncertainty in the number at a particular size may be as large as a factor of 2 for clouds larger than about 5 samples on a side (~4500 km$^2$), based on comparison with a population observed with a much higher resolution dataset, is a useful constraint. This paper does not present any comparisons with models, but based on my prior experience I think it might very well be likely that the differences between simulated and observed size distributions could in fact be larger. Of greater importance, however, may be how robust the scale-dependent properties of the cloud are to the resolution used to detect the clouds. I.e., is an undercount of clouds by as much as a factor of 2 for clouds at 4000 km$^2$ to $10^3$ km$^2$,and an overcount by as much as a factor of 2 for clouds greater than $4*10^4$ km$^2$ lead to a significant difference in the dependence of cloud scale on CAPE and shear, or the scale dependence of the minimum infrared brightness temperature? A complete assessment of this with cloud databases including the co-located reanalysis and microwave datasets for MODIS cloud derived at a variety of averaging scales would be a fairly substantial undertaking and was deemed beyond the scope of this paper. But, as indicated by the reviewer, including this in the uncertainty bounds for the scale-dependent cloud properties when comparing with models would be valuable. This point is now included in the summary section of the paper (p. 12 lines 414-419 of the tracked-changes version of the revised manuscript).

*The discussion of convective lifecycles in the conclusions section was important, but I think this discussion should be introduced much earlier during the "interpretation of database results" parts of the manuscript. For example, lines 183-185 ("The goal...") suggest environment,*

*regional variations, land-ocean differences, but convective life stage also matters here; and, it also matters for cloud top Re differences. Another example: for lines 266-274 (sentences beginning with "These results…"), talking about a size of the cold areas relative to the total size of the cloud and potential relation to mesoscale organization/environments, a recent study (Elsaesser et al. 2022; https://doi.org/10.1029/2021JD035599) showed that convective fraction (which definitely relates to the fraction of cold cloud below 220K discussed here) is very connected to lifecycle, as shown in Figure 2 of that paper. Furthermore, lifecycle also has implications for how Fig. 7 and 8 in this manuscript would be interpreted, because one could imagine only plotting up the maximum sizes along the path of any system as a function of shear/CAPE instead of any instantaneous size. In short, life cycle discussion should be brought in at various parts during discussion of interpretation, instead of first mentioning it in the conclusions.*

This is a valuable suggestion and the discussion has been expanded in sections 2.0 (lines 72-78 in the tracked changes version of the revised manuscript) 2.4 (lines 220-221), 4.1 (lines 352-356), 4.2 (lines 393-395), 5.0 (lines 433-441). Most of this is new in the revised manuscript to address the reviewer's comment that lifecycle effects are important for a complete characterization of the convective processes discussed in this paper, in addition to the discussion in section 5, which was present in the original submission.

*Minor Typos:*

*L285: "…interacting with over convective elements"; should "over" be "other"?*

Yes, this should be "other". It has been corrected.

*L287: "these quantities are certainly not predictive of these quantities" ; rephrase to avoid confusion, and avoid double use of "quantities."*

The second instance of "quantities" has been changed to "cloud scale" to avoid the confusion of the prior wording.

---

## Author Comment (AC2)

Response to reviewer 2

*This papers introduces a dataset focused on atmospheric deep convective cloud built using various satellite observations and a specific object-oriented methodology. Before going further it seems to me this is more a paper relevant for ESSD than for AMT. Indeed the method is not new but its application to MODIS is ! For with this in mind, and so assuming this is more a data paper than a method paper I have the following positive comments.*

*This study aims to provide process oriented observational data to help development and evaluation of numerical atmospheric models and possibly parameterization of deep convection. In this respect the paper is very convincing. The new dataset is a timely addition and a nice consolidation of previously available of existing equivalent geostationnary and polar oribter based datasets. The long span of hte proposed record is indeed a strong asset of the present dataset. The extension to the AMSRE 89GHz information is also a nice new features (that probably needs to be put more forward in the manuscript).*

*The paper is clearly written and well referenced. Indeed the selected illustrations , like the joint PDF of CAPE, shear and size for different regions are very convincing of the possibilities of the dataset for its endeavor. Yet I would welcome a little more cautionnary notes of the use of reanalysis based on assimilation and parameterized GCM for deep convection processes studies.*

We thank the reviewer for their positive reception of our dataset and the goals behind its construction. The desire for additional cautionary discussion about the use of reanalysis in this work, mirrors the comments from reviewer 1. We refer the reviewer to the additional text (also summarized in the response to reviewer 1 where we have expanded upon some of the limitations of reanalysis data for the construction of the dataset and cloud and convection studies using the dataset and the addition of the supplementary figures on the correspondence of simulated convection to observed convection and the new methods we have outlined for quantifying the level of correspondence between the observed IR brightness temperatures and simulated outgoing longwave radiation in the reanalysis data..

---

## Author Response (AR2)

We thank the editors for the careful review of our revised manuscript. In this minor revision we have included the statement of no competing interests above the acknowledgments section and we have fixed the figure numbering and formatting in the caption for the first figure in the supplement.

---

## Author Response (AR3)

Response to editor for production manuscript upload:

We thank the editor for managing the review of this manuscript. Below is a list of changes made to the manuscript in advance of uploading the production version of the manuscript.

1) A citation to the dataset documented in the paper is referenced at the end of the first paragraph in section 2.5 (line 226) and the reference has been added to the list of references.
2) The date ranges for the Terra MODIS and Aqua MODIS datasets in line 231 have been adjusted slightly to be consistent with the first version of the dataset being hosted by NASA. These date ranges will be expanded in a reprocessing update of the dataset planned in the next few months.
3) "Code availability", "Data availability" and "Author contribution" sections have been added between the conclusion section and the "Competing interests" sections. The content of the "Data availability" section has been amended to list only the permanent DOIs assigned to the deep cloud database.